# Structural basis for functional interactions in dimers of SLC26 transporters

Yung-Ning Chang [1,7], Eva A. Jaumann[2,7], Katrin Reichel[3,7], Julia Hartmann[4], Dominik Oliver [4,5], Gerhard Hummer [3,6], Benesh Joseph [2,6] & Eric R. Geertsma [1]

The SLC26 family of transporters maintains anion equilibria in all kingdoms of life. The family shares a $7+7$ transmembrane segments inverted repeat architecture with the SLC4 and SLC23 families, but holds a regulatory STAS domain in addition. While the only experimental SLC26 structure is monomeric, SLC26 proteins form structural and functional dimers in the lipid membrane. Here we resolve the structure of an SLC26 dimer embedded in a lipid membrane and characterize its functional relevance by combining PELDOR/DEER distance measurements and biochemical studies with MD simulations and spin-label ensemble refinement. Our structural model reveals a unique interface different from the SLC4 and SLC23 families. The functionally relevant STAS domain is no prerequisite for dimerization. Characterization of heterodimers indicates that protomers in the dimer functionally interact. The combined structural and functional data define the framework for a mechanistic understanding of functional cooperativity in SLC26 dimers.

[1] Institute of Biochemistry, Biocenter, Goethe University Frankfurt, Max-von-Laue Str. 9, 60438 Frankfurt am Main, Germany. [2] Institute of Physical and Theoretical Chemistry, Goethe University Frankfurt, Max-von-Laue Str. 7, 60438 Frankfurt am Main, Germany. [3] Department of Theoretical Biophysics, Max Planck Institute of Biophysics, Max-von-Laue Str. 3, 60438 Frankfurt am Main, Germany. [4] Department of Neurophysiology, Institute of Physiology and Pathophysiology, Philipps University, 35037 Marburg, Germany. [5] DFG Research Training Group, Membrane Plasticity in Tissue Development and Remodeling, Philipps University, GRK 2213 Philipps, Germany. [6] Institute of Biophysics, Goethe University Frankfurt, Max-von-Laue Str. 1, 60438 Frankfurt am Main, Germany. [7] These authors contributed equally: Yung-Ning Chang, Eva A. Jaumann, Katrin Reichel, Correspondence and requests for materials should be addressed to G.H. (email: gerhard.hummer@biophys.mpg.de) or to B.J. (email: joseph@biophysik.uni-frankfurt.de) or to E.R.G. (email: geertsma@em.uni-frankfurt.de)

The solute carrier family 26 (SLC26), also known as the sulfate permease (SulP) family, facilitates the transport of a broad variety of organic and inorganic anions[1]. Members of this family are found in all kingdoms of life and operate predominantly as secondary transporters (symporters and exchangers)[2–4]. As an exception, prestin (SLC26A5) functions as a voltage-sensitive motor protein that evokes robust length changes in outer hair cells and thereby contributes to cochlear amplification[5,6]. The relevance of the SLC26 family in maintaining anion equilibria is underlined by the causative role of mammalian SLC26 proteins in diseases such as congenital chloride diarrhea[7] and cytotoxic brain edema[8].

SLC26 proteins are composed of a membrane-inserted transport domain and a carboxy-terminal cytoplasmic STAS (sulfate transporter and anti-sigma factor antagonist) domain. The SLC26–STAS domain is relevant for intracellular trafficking[9,10] and protein–protein interactions[11–13]. Its deletion impairs substrate transport by the membrane domain[4,10,14]. The crystal structure of SLC26Dg, a prokaryotic SLC26 protein from *Deinococcus geothermalis*, revealed a spatially separated membrane and STAS domain[4]. The SLC26Dg membrane domain holds two intertwined inverted repeats of seven transmembrane segments (TMs). Despite a poor sequence homology, the SLC26 family shares this 7-TM-inverted repeat (7TMIR) architecture with the SLC4 and SLC23 families that transport bicarbonate and nucleobases plus vitamin C, respectively[15–22]. The 14 TMs are arranged in two subdomains: a compact core domain that holds the substrate binding site as inferred from the location of the nucleobases in the SLC23 crystal structures[20–22], and an elongated gate domain that shields one side of the core domain. A mounting body of evidence[16,19,22,23] suggests that these proteins operate based on an elevator alternating-access mode of transport[24] involving a rigid-body translation-rotation of the core domain with respect to the gate domain.

Dimeric states have been previously observed for pro- and eukaryotic members of the SLC4[25–27], SLC23[28], and SLC26[4,29–32] families. Recent structures subsequently confirmed this oligomeric state for SLC4[17–19] and SLC23[21,22] proteins and indicated that in both families the gate domains form the main interaction surface between protomers, though each family appears to hold a distinct dimer interface. As the crystal structure of SLC26Dg captured the protein in a monomeric state, the mode of interaction between SLC26 protomers has remained elusive. Interestingly, the protomers within the SLC26 dimer have been found to interact functionally[29,33] despite the presence of a complete translocation path in each individual protomer. Here, we provide structural and mechanistic insights in the allosteric interactions between SLC26 protomers. We integrated pulsed electron–electron double resonance (PELDOR, also known as double electron–electron resonance) distance measurements and in vitro transport studies with structural modeling and refinement using molecular dynamics (MD) simulation to determine the architecture of the membrane-embedded SLC26 dimer and characterize its functional relevance.

## Results
### Resolving the SLC26Dg dimer interface in the lipid membrane.
To define the SLC26Dg dimer interface, we used interspin distance constraints derived from PELDOR experiments. As SLC26Dg is monomeric in its detergent-solubilized state, we reconstituted spin-labeled protomers in lipid membranes to assure dimer formation[4]. As the gate domain in the SLC4[17–19] and SLC23[21,22] families establishes the main protomer–protomer contacts in the membrane (Fig. 1a, b), we engineered spin labels at 13 different positions on the termini of gate domain helices in

SLC26Dg. One additional central position in the core domain (TM8) was also selected (Fig. 1c). By site-directed modification of single-cysteine mutants, we efficiently introduced the probe 1-oxyl-2,2,5,5-tetramethylpyrroline-3-methyl-methanethiosulfonate (MTSSL)[34] (Supplementary Table 1, Supplementary Fig. 1). Size-exclusion chromatography and transport assays in proteoliposomes demonstrated that all mutants were folded well and active (Supplementary Fig. 2).

Systematic analysis of all positions led to the identification of three labeled positions, K353R1, V367R1, and L385R1, that gave well-defined interspin distance distributions centering around $4.4 \pm 0.2$, $3.9 \pm 0.3$, and $1.8 \pm 0.1$ nm, respectively (Fig. 2b–d). These positions are located in TM13 and TM14 and place this region in close proximity to the center of the SLC26Dg dimer interface. This particular dimer arrangement combined with the short phase memory time ($T_M$) of the spins in membranes (Supplementary Fig. 3) did not allow to accurately determine the long interspin distances between other helices (Supplementary Fig. 4). An exponential decay was observed for the PELDOR measurement of the detergent-solubilized protein (Fig. 2a), supporting the notion that the identified region is part of the native SLC26Dg dimer interface formed in the lipid membrane. Given the spin-labeling efficiencies of 70–100%, the obtained modulation depths of the PELDOR time traces are in the range expected for a dimer (Supplementary Table 1), suggesting that the majority of the protomers in the membrane is part of a dimer.

### Structural model of the SLC26Dg membrane dimer interface.
On the basis of the PELDOR data and the SLC26Dg crystal structure, we constructed a dimer model. First, to obtain an equilibrated structure for rigid-body docking, monomeric SLC26Dg without its carboxy-terminal STAS domain was embedded in a palmitoyl-oleoyl-phosphatidylcholine (POPC) bilayer and submitted to 1 μs of MD simulations. We observed considerable flexibility of the gate domain in comparison with the core domain (Fig. 2e). In particular, TM13 and TM14 exhibited significant motions in the monomer, in line with their suspected involvement in the membrane dimer interface. Owing to the observed flexibility, we used several relaxed monomer conformations obtained at 110 ns intervals of MD for docking. For each conformation, a rigid-body search restricted by C2 symmetry with an axis normal to the membrane was performed and the rotation angle that showed the best overall fit with the PELDOR data was determined. Using this approach, we identified a candidate dimer structure based on a monomer conformation observed at 440 ns of MD and a polar plane angle of $210 \pm 5°$ (Supplementary Fig. 5). An alternative rigid-body docking approach guided by the inferred distance distributions resulted in a very similar dimer model (Supplementary Fig. 6). The initial C2 symmetric MD dimer was then relaxed by additional MD simulation. The forward-calculated PELDOR traces for the relaxed dimer, after gentle spin-label rotamer refinement[35], are in excellent agreement with the experimental background-corrected time-domain data (Fig. 2b–d, left panels, Supplementary Fig. 7). The structure of this SLC26Dg membrane domain dimer model is shown in Fig. 3. Simulations performed on this model for two additional positions, V129R1 (TM5) and L248R1 (TM8), agree with the experimental data as well (Supplementary Fig. 4a). For the other nine positions in the gate domain, simulations predict mean interspin distances in the range of 6.3–10.6 nm, which could not be accurately determined owing to the short $T_M$ (Supplementary Fig. 4b and Supplementary Fig. 3).

The model of the SLC26Dg dimer displays a protomer–protomer membrane interface that is remarkably

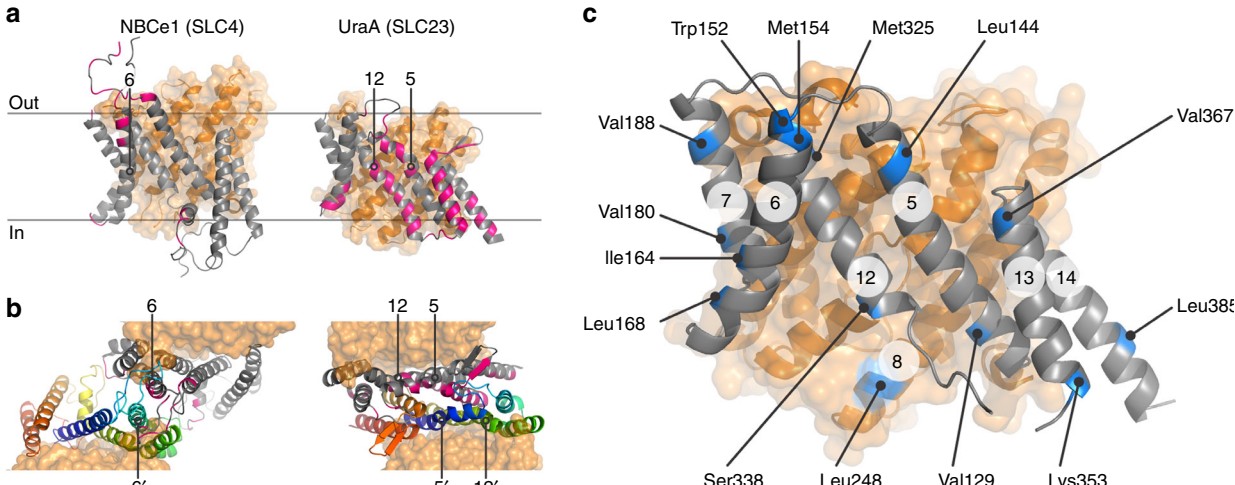

**Fig. 1** Dimer interfaces in 7TMIR proteins. **a** Side view of the membrane domains of NBCe1 (PDB: 6CAA) and UraA (PDB: 5XLS). Core and gate domain are colored orange and gray, respectively, with residues within 4 Å of the opposing protomer in pink. **b** Top views of the dimeric arrangements of NBCe1 and UraA. For each dimer, the gate domain of one of the protomers follows a rainbow coloring scheme (blue-to-red for N-to-C direction). TMs central in the respective dimers are numbered. **c** Side view of the membrane domain of SLC26Dg (PDB: 5DA0). Residues mutated to cysteine for site-directed spin labeling are colored blue. The circled numbers indicate the respective TMs

different from the membrane interfaces observed for the SLC4 and SLC23 families, both in its location and in its size[17–19,21,22]. Whereas the membrane dimer interfaces of SLC4 and SLC23 proteins center around TM6, and TM5 plus TM12, respectively, the midpoint of the SLC26Dg dimer is TM14. Furthermore, although the membrane dimer interface of SLC4 and SLC23 proteins involves extensive interactions covering large fractions of the exposed membrane surface of their gate domains, the membrane interface of SLC26Dg is relatively small. Also, in comparison with other oligomeric membrane proteins, the surface buried by dimerization of the membrane domain is modest[36]. This observation agrees with the complete absence of dimerization in detergent and suggests that other factors, such as subunit-bridging lipids or the cytoplasmic STAS domain may contribute to the stabilization of the dimeric state.

**STAS domain affects central regions in the dimer**. The cytoplasmic STAS domain is one of the major structural constituents that distinguishes the SLC26 family from the SLC4 and SLC23 families, which do not hold carboxy-terminal domains[16]. Although deletion of the STAS domain compromises the transport capacity of the SLC26Dg membrane domain, the structure of the membrane domain is not altered[4]. As the STAS domain immediately follows the central TM14, we further determined to what extent the STAS domain contributes to the dimer interface.

As evidenced from the PELDOR time trace for L385R1 in SLC26Dg$^{\Delta STAS}$, deletion of the STAS domain did not affect the ability of the membrane domain to form dimers (Supplementary Fig. 8). STAS domain deletion resulted in a small increase in the mean L385R1 distance from $1.8 \pm 0.1$ to $2.1 \pm 0.1$ nm, that, given the narrow distance distribution, rather suggests a rearrangement of the MTSSL rotamers than a physical separation of the protomers. The complete disappearance of oscillations in the primary PELDOR data of SLC26Dg$^{\Delta STAS}$-K353R1 and -V367R1 in TM13 suggests that either similar rearrangements of spin-label rotamers or an increased flexibility at these positions may underlie these changes (Supplementary Fig. 8). The latter could not be confirmed owing to the limited time window of the dipolar evolution. Thus, although deletion of the STAS domain appears to affect the environment around the spin labels in TM13 and

TM14, the STAS domain itself is not a prerequisite for dimerization.

**SLC26Dg dimer interface represents the SLC26 family**. To further validate the SLC26Dg membrane dimer model and determine to what extent it represents the SLC26 family in general, we used oxidative cross-linking in biological membranes. Owing to its central position, we focused on TM14 (Fig. 3b). Oxidative cross-linking of single-cysteine variants at several positions in TM14 of SLC26Dg, fused to superfolder green fluorescent protein (GFP) to facilitate detection, leads to the appearance of a band with lower electrophoretic mobility (Fig. 4a). We assign this band to SLC26Dg homodimers because an identical anomalous shift was observed on cross-linking in proteoliposomes (Supplementary Fig. 9). Cross-links were observed for residues located at both ends of TM14, but not for residues facing the interior of the bilayer in line with a general lower reactivity of cysteines at this position[37–39]. The ability of cysteine residues in TM14 of SLC26Dg to form a disulfide bond with the opposing protomer further validates our SLC26Dg dimer model (Fig. 4b).

Although the known dimer interfaces between SLC4 and SLC23 families differ greatly, a high degree of similarity is observed between members of the same family[17–19,21,22]. This suggests that the dimer interfaces for this fold are specific to a family. To test this, we used the same TM14 cross-linking approach on SLC26 proteins from *Sulfitobacter indolifex* and *Rattus norvegicus*, which hold 23% and 21% sequence identity to SLC26Dg, respectively (Supplementary Fig. 10, 11). For both proteins, we observed the formation of TM14 disulfide cross-links between protomers, which provides evidence that the membrane dimer interface may be very similar, if not conserved, throughout the SLC26 family.

**Functional relevance of the SLC26Dg dimer**. The observation of a structural SLC26Dg dimer led us to ask whether this oligomeric state is important for function. As both protomers have independent binding sites and non-overlapping translocation paths, the relevance of the dimeric state is not evident. Functional interactions between protomers in oligomeric proteins can be revealed by mixing protomers with different functional

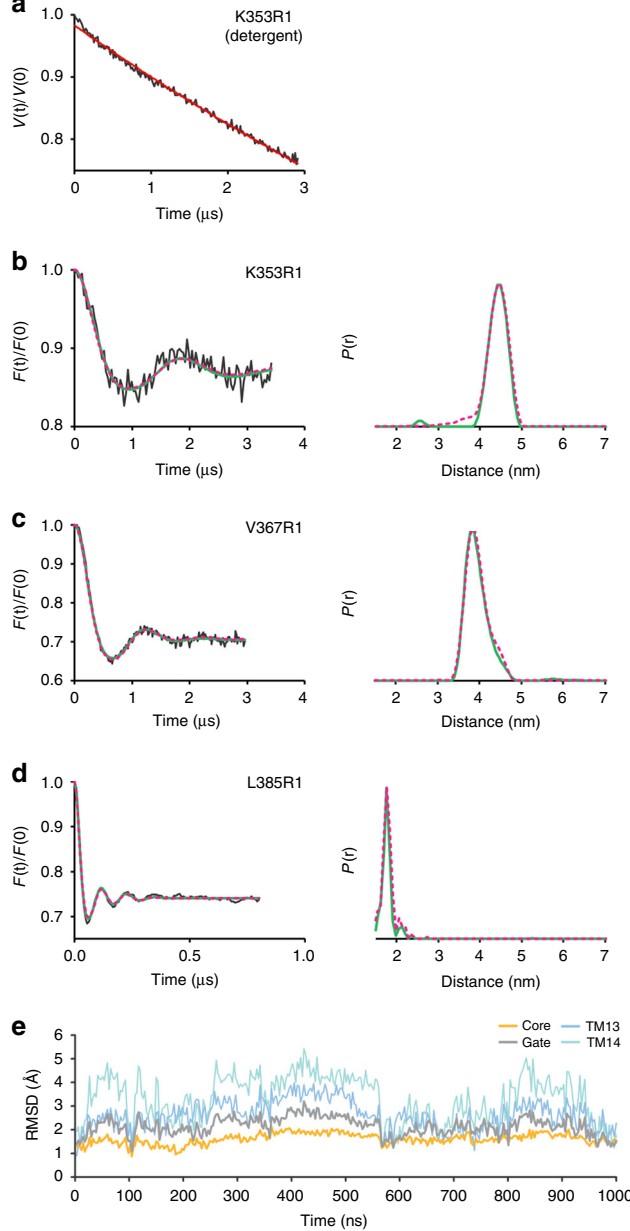

**Fig. 2** Interspin distances in the SLC26Dg dimer. **a** Primary PELDOR data of detergent-solubilized K353R1. **b**–**d** Left panels: background-corrected PELDOR time traces for membrane-reconstituted K353R1, V367R1, and L385R1 (black traces), overlaid with the fit from Tikhonov regularization (green), and forward-calculated PELDOR time traces from BioEn spin-label rotamer refinement of the MD simulation model (magenta, dashed; $\theta = 10$). Right panels: distance distributions obtained by Tikhonov regularization (green), overlaid with the distance distributions resulting from BioEn analysis of the MD simulation model (magenta, dashed). Original PELDOR data in Supplementary Fig. 2. **e** $C_\alpha$-atom root mean squared distance (RMSD) values of the core, gate, TM13, and 14 relative to the monomer crystal structure as a function of MD time (1 μs)

characteristics and analyzing the resulting hetero-oligomers. We opted to create an inactive variant by locking the protein in the inward-facing conformation using disulfide cross-linking. Based on the crystal structure, we selected Ile-45 on the extracellular side of TM1 (core) and Ala-142 in TM5 (gate) as most suited positions concerning cross-linking efficiency and ability to lock the protein (Fig. 5a). Oxidative cross-linking of SLC26Dg-CL-

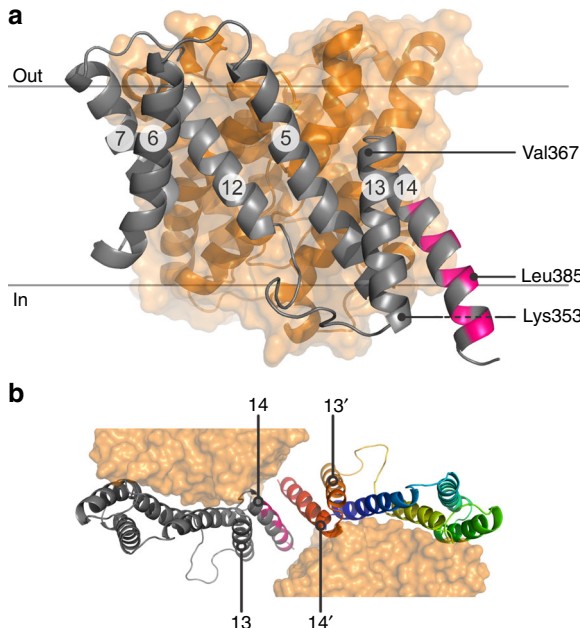

**Fig. 3** Model of the SLC26Dg dimer interface. **a** Side view of the SLC26Dg membrane domain in the same orientation as Fig. 1a. Core and gate domain are colored orange and gray, respectively, with residues within 4 Å of the opposing protomer in pink. **b** Top views of the dimeric arrangement of SLC26Dg. The gate domain of one of the protomers follows a rainbow coloring scheme (blue-to-red for N-to-C direction)

I45C/A142C, hereafter named SLC26Dg-IL (inward-locked), resulted in a nearly complete shift in the electrophoretic mobility of the protein that could be restored by the addition of the reductant dithiothreitol (DTT) (Fig. 5b). Likewise, fumarate transport of the cross-linked mutant in proteoliposomes was close to background activity, but could be fully recovered to wildtype activity by the addition of DTT, indicating that the protein was well-folded and reconstituted (Fig. 5c).

As SLC26Dg is monomeric in detergent and dimerizes only after reconstitution in the lipid membrane, we achieved stochastic formation of heterodimers as demonstrated by the decreased TM14 cross-linking upon the addition of SLC26Dg-IL in a control experiment (Supplementary Fig. 12). Interestingly, the initial transport rates of proteoliposomes holding different ratios of wildtype and SLC26Dg-IL followed a positive quadratic relationship (Fig. 5d). The activity of the heterodimers exceeded the expected values for independent functioning of protomers, which is half the sum of the activities of the wildtype and SLC26Dg-IL homodimers (Fig. 5d, straight line). In fact, in the most parsimonious model for the quadratic dependence of the activity on the mixing ratio, only the SLC26Dg-IL homodimer is inactive and all other dimers have the same activity. This could imply that either only one protomer is active in a dimer or the activity of a wildtype protomer is doubled when paired with an inward-locked protomer. In any case, the robust coupling evident in this transport activity data is a strong indication that dimerization is functionally relevant.

## Discussion

The structural model of the SLC26Dg membrane domain dimer interface, based on electron paramagnetic resonance (EPR) measurements on membrane-reconstituted protein and validated by cysteine cross-linking in biological membranes, places TM14 of the gate domain at the center of the SLC26Dg dimer. Further cross-linking studies on additional prokaryotic and mammalian

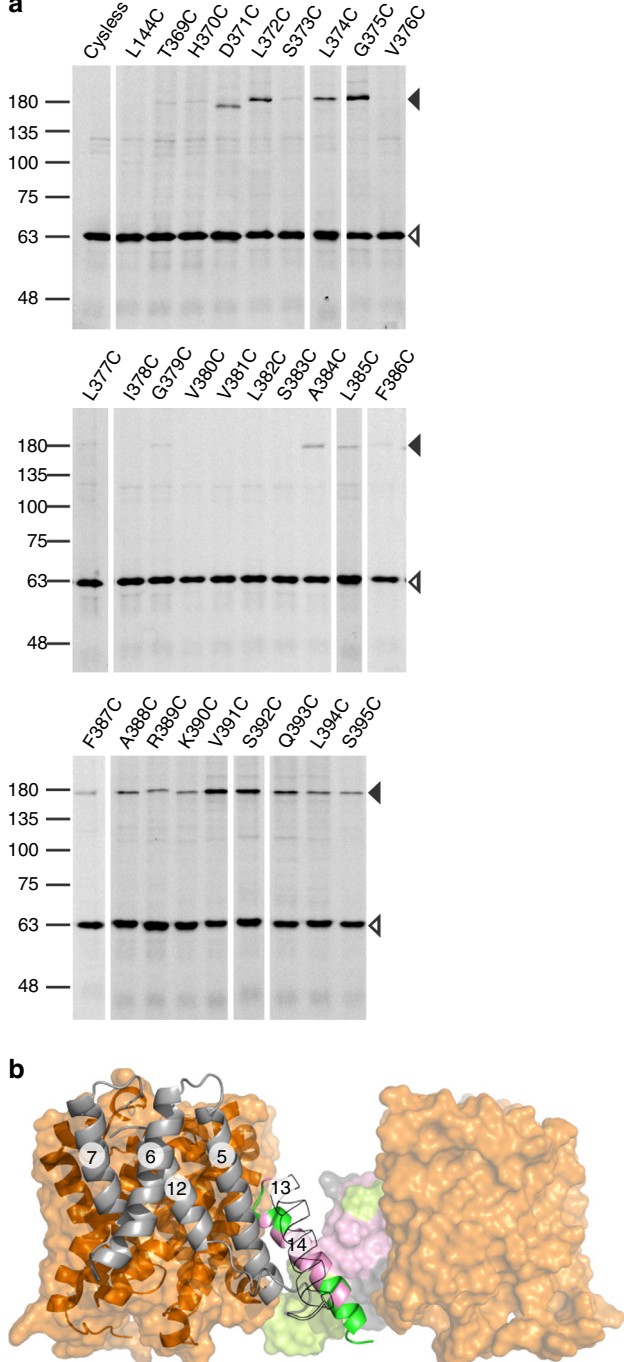

**Fig. 4** Oxidative cysteine cross-linking between TM14 of SLC26Dg. **a** In gel GFP fluorescence analysis of disrupted *E. coli* cells expressing single-cysteine variants of SLC26Dg fused to superfolder GFP. Following oxidative cross-linking, samples were analyzed by non-reducing SDS-PAGE. Cysteine-free SLC26Dg (cysless) and L144C (TM5) represent negative controls. Black and white arrows indicate dimeric and monomeric SLC26Dg. Source data are provided as a Source Data file. **b** Side view of the SLC26Dg dimer model. Core and gate domain are colored orange and gray, respectively. Positions in TM14 susceptible to cross-linking are colored in green, non-susceptible residues are colored in pink. The gate domain of the right protomer is depicted in surface representation. TM13 of the left protomer is contoured. The circled numbers indicate the respective TMs

homologs suggest that this interface might be evolutionary conserved in the SLC26 family. Nevertheless, amino-acid sequence alignments of TM14 show no conserved features between or even within prokaryotic and mammalian SLC26 proteins, other than a GxxxG-like motif toward the extracellular side of TM14 (Supplementary Fig. 13). Though this motif often mediates dimerization in single-pass membrane proteins[40], its role in multi-pass membrane proteins lies more likely in folding of the protomers[41]. Besides, this specific region of TM14 is not directly involved in protomer–protomer interactions in our model. Specificity of the SLC26 protomer–protomer interaction may instead arise from general complementarity of the interacting surfaces combined with other, potentially conserved features such as interfacial lipids (*vide infra*). The central position of the gate domain in the SLC26 membrane dimer interface corresponds well with the SLC4 and SLC23 families in which the gate domains also form the major contacts between the membrane domains[17–19,21,22]. However, although the dimer interfaces seem conserved within a family, the regions involved in the protomer–protomer contacts seem to differ greatly among the three families.

It appears likely that these different dimeric arrangements represent stable constellations between which the protomers do not alternate during transport. The structures of dimeric SLC4 and SLC23 proteins in different conformations[17–19,21,22] have identical contact surfaces within each family. This is further supported by repeat-swap homology modeling of AE1, which indicated that no changes in the dimerization interface were required during the transition from the outward-facing structure to the inward-facing model[23]. Transitions between interfaces seem further unlikely owing to the requirement for significant rearrangements in structural elements, such as the cytoplasmic region following TM12 in SLC4A1 and SLC4A4[17,18], which also appears stable and blocking alternative interfaces in SLC26Dg (Supplementary Fig. 14). Finally, our PELDOR data on SLC26Dg, especially the well-defined distance distributions for the interface region, strongly disagrees with a dynamic interface. A stable oligomer interface is in line with other observations on unrelated elevator proteins[42,43].

The buried surface resulting from dimerization in the SLC26Dg membrane domain amounts to ~ 350 Å$^2$, which is small, not only in comparison with the SLC4 and SLC23 family whose membrane interfaces measure ~ 1000 Å$^2$ and ~ 2000 Å$^2$ [44], respectively, but also in relation to other oligomeric membrane proteins[36]. It is likely that additional extrinsic factors contribute to extend and stabilize the SLC26 membrane dimer interface, e.g., interfacial lipids that were recently reported to stabilize an SLC23 dimer[45] and other oligomeric membrane proteins[36]. In this respect, the STAS domain appears to be relevant as well. The short linker region connecting TM14 and the STAS domain implies its close proximity to the membrane dimer interface. In addition, the STAS domain affects TM13 and TM14 at the center of the dimer interface (Supplementary Fig. 8). Given that isolated SLC26–STAS domains do not appear to form dimers[13,46,47], we expect the STAS-mediated effect on the gate domain to result from a direct interaction between the STAS domain and the membrane domain. This interaction may also form the basis for the enhanced transport rates observed in the presence of the STAS domain[4,10,14].

All 7TMIR proteins form structural dimers in the membrane, but the general relevance of this oligomeric state for their function is not clear. The available structures of the SLC4, SLC23, and SLC26 proteins all indicate that the complete substrate translocation path is contained within one protomer. This is supported

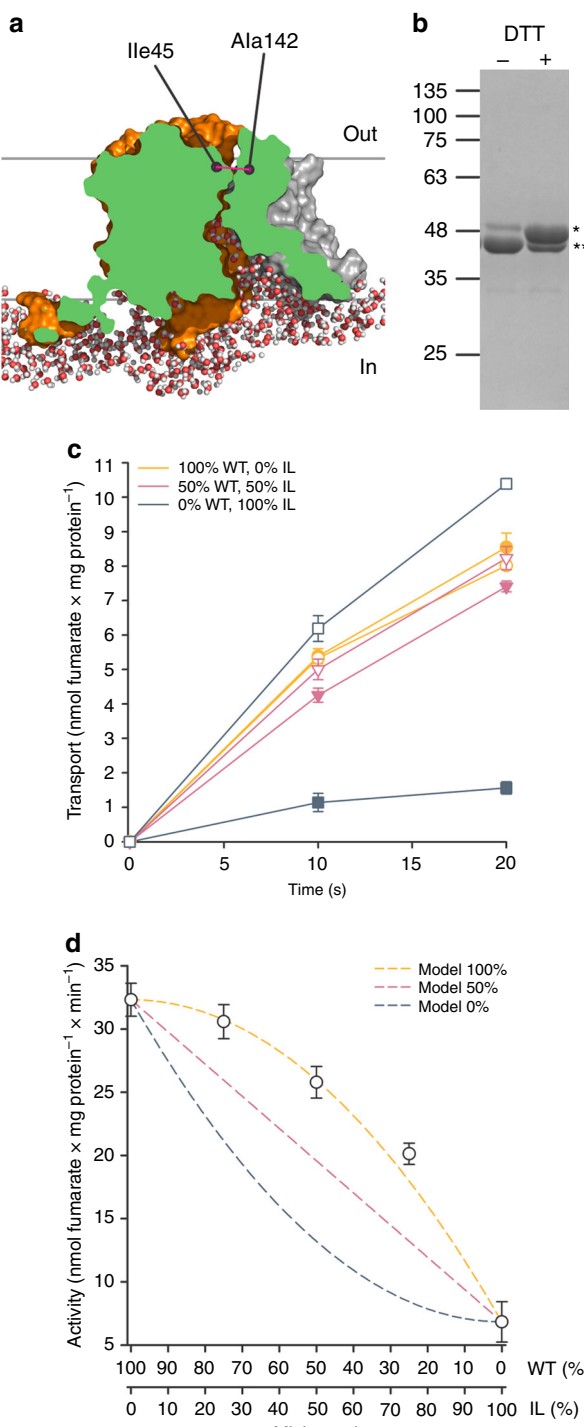

**Fig. 5** Generation and functional characterization of SLC26Dg-IL. **a** Surface representation of MD-simulated SLC26Dg clipped through the funnel toward the putative substrate-binding site. Cytoplasmic water molecules in a ~10 Å slab at the clipping plane are shown. Ile-45 and Ala-142 indicate the relative position of the cysteine mutants in the core (orange) and gate (gray) domain, respectively. **b** SDS-PAGE analysis of purified and cross-linked SLC26Dg-IL monomers in the absence and presence of DTT. Single and double stars indicate not-cross-linked and cross-linked protein, respectively. **c** Functional characterization of membrane-reconstituted and cross-linked SLC26Dg-IL (dark blue), wildtype SLC26Dg (orange), and both proteins mixed in equal ratio's (pink). Closed and open symbols indicate the absence and presence of a pre-incubation step with DTT. **d** Initial transport rates of membrane-reconstituted and cross-linked samples containing wildtype and SLC26Dg-IL mixed in different ratio's. Dark blue, pink, and orange dashed curves indicate the anticipated curves assuming an activity of the heterodimers corresponding to 0, 50, and 100% of the wildtype homodimers. These models were calculated assuming stochastic dimer formation (e.g., mixing WT:IL protomers in a 50:50 ratio results in 25% WT–WT, 50% WT–IL, and 25% IL–IL dimers) and specific transport activities of 32.3 or 6.8 nmol fumarate per mg WT or IL homodimer per min, respectively, and heterodimer activities corresponding to 0, 50, or 100% of WT homodimer. Data points represent mean and standard deviations of three technical replicates. Source data are provided as a Source Data file

although the inferred specific activity of the heterodimers of NBCe1 (50% active) suggests that the protomers can operate independently, the apparent negative and positive dominance observed for UapA, and UraA plus SLC26Dg, respectively, indicates that the dimeric state may have a functional role as well. This notion is further supported by studies on rat prestin (SLC26A5) heterodimers composed of protomers that in the context of a homodimer hold a very different voltage-dependence of their non-linear capacitance. For these heterodimers an intermediate phenotype was observed, suggesting a strong co-operative interaction in which the two protomers jointly determine the voltage-dependence of the conformational changes[29].

Though the mechanistic basis for functional interactions in 7TMIR dimers is currently unclear, important insights were obtained from the characterization of monomeric 7TMIR proteins. Monomeric variants of UraA, generated by the introduction of bulky residues at the dimer interface, bind substrate with wildtype affinities and are thus expected to be well-folded, but are incapable of facilitating transport[22]. In our study, we characterized the transport properties of individual protomers as well, but in the context of a dimer. These SLC26Dg-WT protomers, embedded in WT–IL heterodimers, are fully active. In fact, the WT–IL heterodimers have the same activity as wildtype homodimers. Together these observations highlight the relevance of the interaction between opposing gate domains for facilitating transport. This interaction may stabilize an essential conformation of the gate domain required for transport, as suggested previously[22] and in line with our observation that the transport-incompetent SLC26Dg$^{\Delta STAS}$ undergoes small rearrangements in the gate domain. Alternatively, the gate–gate domain interaction may provide a stable membrane-embedded scaffold that enables the vertical translation of the core domain and its anticipated deformation of the bilayer. In this context, the inward-locked SLC26Dg protomer may serve as an extended scaffold that fixates the gate domain even better in the membrane, providing a rational for the apparent increase in transport rate observed for the wildtype protomer in the heterodimer. Though the similar transport rates of wildtype homodimers and WT–IL heterodimers may also imply that only one protomer is active in the

by the recessive inheritance mode of SLC26-linked diseases[29,48] and further confirmed by the functional characterization of heterodimers composed of a wildtype and an inactive mutant protomer. Most of these heterodimers were found to be active for NBCe1[49] (SLC4), UraA[22] (SLC23), and SLC26Dg (SLC26, this work), though for UapA (SLC23), inactive heterodimers were observed as well[21]. With the exception of SLC26Dg, these studies were carried out in the context of whole cells, employed different mutations that interfered in diverse ways with substrate transport, and, in case of NBCe1 and UraA, involved the use of concatemeric constructs. This diversity in experimental approaches makes it difficult to precisely compare these data. Nevertheless,

SLC26Dg dimer, the latter appears in conflict with the intermediate non-linear capacitance observed for rat prestin heterodimers[29]. Additional structures of dimeric 7TMIR proteins in multiple states will be required to further pinpoint the role of the gate domain.

Understanding the transport mechanism of 7TMIR proteins requires that proteins are not studied only as individual protomers, but also in the context of the dimer, their functionally relevant oligomeric state. Structures of SLC4 and SLC23 proteins have provided exceptional insight into protomer interactions by providing snapshots of dimeric constellations, but the structure of a dimeric SLC26 protein has been elusive. Here, we have determined the architecture of dimeric, membrane-embedded SLC26Dg using an integrated structural biology approach. The SLC26 dimer interface is unique and distinguishes itself from SLC4 and SLC23 proteins. We have demonstrated that the interface is not dynamic, and that the carboxy-terminal STAS domain, although not required for dimerization, affects regions central in the dimer. Finally, our heterodimer studies have underlined the functional significance of the dimer. Together these structural, dynamic, and functional characterizations provide the framework for further studies on the SLC26 family and offer mechanistic insights that may extend to other elevator proteins as well.

## Methods

**Site-specific mutagenesis of SLC26 transporters**. Cysteine residues were introduced into pINITcat-SLC26Dg by Quikchange mutagenesis or a two-step PCR method (mega-primer approach; primer sequences in Supplementary Data 1). Sequence-validated pINITcat-SLC26Dg variants were subsequently subcloned into pBXC3GH by FX cloning[50] for protein expression and purification.

**Protein expression and purification**. *Escherichia coli* MC1061 (ATCC 53338) containing pBXC3GH-SLC26Dg or the variants was cultivated in 9 L TB/ampicillin in a fermenter (Bioengineering). Cells were grown at 37 °C until an $OD_{600} \approx 2$ was reached, after which the temperature was gradually decreased to 25 °C over the course of 1 h. Expression was induced by the addition of 0.005% (w/v) L-arabinose and continued for 16 h. Cell pellets were resuspended in 50 mM potassium phosphate (KPi), pH 7.5, 150 mM NaCl, and 1 mM $MgSO_4$ and incubated for 1 h at 4 °C in the presence of 1 mg/mL lysozyme and traces of DNase I before disruption with an APV Gaulin/Manton homogenizer. The lysate was cleared by low-spin centrifugation, and membrane vesicles were obtained by ultracentrifugation. Vesicles were resuspended to 0.5 g/mL in 50 mM KPi, pH 7.5, 150 mM NaCl and 10% glycerol (buffer A). All subsequent steps were carried out at 4 °C. Membrane proteins were extracted for 1 h at a concentration of 0.1 g/mL buffer A supplemented with 1–1.5% (w/v) *n*-decyl-β-maltoside (DM, Glycon). Solubilized SLC26Dg was purified by immobilized metal affinity chromatography (IMAC). Target protein was immobilized on Ni-NTA resin and impurities were removed with 20 column volumes (CV) washing with 20 mM HEPES, pH 7.5, 150 mM NaCl (buffer B) supplemented with 50 mM imidazole, pH 7.5 and 0.2% DM. Protein was eluted with buffer B containing 300 mM imidazole and cleaved with HRV 3 C protease during dialysis against buffer B without imidazole. Histidine-tagged GFP and protease were removed by IMAC, and cleaved protein was concentrated and subjected to size-exclusion chromatography (SEC) on a Superdex 200 Increase 10/300 column (GE Healthcare) equilibrated with 20 mM HEPES, pH 7.5, 150 mM NaCl, and 0.2% DM (buffer C).

**Site-directed spin labeling of SLC26Dg cysteine mutants**. Cultivation and isolation of membrane vesicles were essentially performed as detailed above, but buffers for resuspending cells and membrane vesicles were supplemented with 3 mM, and 1 mM DTT, respectively. IMAC purification was conducted in the same way as described in the previous section but 5 mM 2-mercaptoethanol was included in all purification buffer to preserve the reduced state of the cysteine residues. Peak fractions from SEC purification were pooled and 2-mercaptoethanol was removed with Econo-Pac 10DG desalting column (Bio-rad), which was pre-equilibrated with buffer C. The concentration of eluted protein was adjusted to 7.5 μM with buffer C. The labeling of cysteine residue was initiated by stepwise addition of 100 mM MTSSL spin label (in dimethyl sulfoxide, Toronto Research Chemicals) in the protein solution to a final concentration of 300 μM and incubated at room temperature for 45 min with gentle agitation. The spin-labeled protein was further concentrated and free label was removed using Micro Bio-Spin 6 Chromatography Columns (Bio-rad) pre-equilibrated with buffer C.

**Membrane reconstitution of SLC26Dg**. Proteoliposomes were prepared using the detergent-doped liposomes method[4,51]. Dry pellets of L-α-phosphatidylcholine (derived from soybean, Sigma) were dissolved in chloroform, dried in a rotary evaporator, resuspended to 20 mg/ml and sonicated in buffer containing 50 mM KPi, pH 7.5. After three freeze–thaw cycles, large unilamellar vesicles were prepared by extrusion through a polycarbonate filter with pore diameters of 400 nm. Liposomes were diluted to 4 mg/ml and destabilized beyond $R_{sat}$ with Triton X-100. SEC-purified SLC26Dg in 0.2% DM was added to the liposomes at a weight ratio of 1:50 (protein/lipid) for transport assays or 1:20 (protein/lipid) for PELDOR measurements, and detergent was subsequently removed by the addition of Biobeads. For radioisotope transport assays, proteoliposomes were harvested by centrifugation for 1.5 h at 250,000 × g and resuspended in 50 mM sodium phosphate (NaPi), pH 7.5, 2 mM $MgSO_4$ to a lipid concentration of 20 mg/ml. After three freeze–thaw cycles, proteoliposomes were stored in liquid nitrogen until analysis. For PELDOR measurements, proteoliposomes were harvested by centrifugation for 20 min at 250,000 × g and resuspended in 50 mM KPi, pH 8.0 to a final spin concentration of 80–130 μM.

**PELDOR EPR**. All the PELDOR experiments were performed at Q-band frequencies (33.7 GHz) using a Bruker E580 spectrometer equipped with an EN 5170 D2 cavity, 150 W traveling-wave tube (Applied Systems Engineering Inc.) microwave amplifier, and an ELEXSYS SuperQ-FT accessory unit. The temperature was kept at 50 K using an ITC 502 temperature control unit (Oxford Instruments) and a continuous-flow helium cryostat (CF935, Oxford Instruments). For all samples, 20 % (v/v) deuterated glycerol was added. For measurement, a 10 μL sample was transferred into a 1.6 mm outer diameter quartz EPR tubes (Suprasil, Wilmad LabGlass) and immediately frozen in liquid nitrogen. The dead-time free four-pulse PELDOR sequence with a phase-cycled π/2-pulse was used[52,53]. Typical pulse lengths were 22 ns (π/2 and π) for the observer pulses and 12 ns (π) for the pump pulse. The delay between the first and second observer pulse was increased by 16 ns for eight steps to average deuterium modulations. The frequency of the pump pulse was set to the maximum of the echo-detected field swept spectrum to obtain maximum inversion efficiency. The observer frequency was set 70 MHz lower. Distance distributions were determined using DeerAnalysis[54]. The normalized primary PELDOR data $V(t)/V(0)$ were processed to remove the intermolecular contribution and the resulting form factors $F(t)/F(0)$ were fitted with a model-free Tikhonov regularization to determine the distance distributions. The MATLAB-based MMM[55] software was used for simulation of interspin distances on the form factor-based dimer model.

**CW EPR**. Continuous wave (CW)-EPR spectra were recorded to determine the spin-labeling efficiency. The spectra were recorded at a Bruker ELEXSYS E500 spectrometer (9.4 GHz) at room temperature with the following parameter settings: microwave power of 2.00 mW, modulation amplitude of 0.15 mT, and modulation frequency of 100 KHz.

**MD simulation of the SLC26Dg monomer**. The crystal structure of the membrane domain of SLC26Dg monomer (PDB: 5DA0)[4] was used in all-atom explicit solvent MD simulation for equilibration and to uncover structural flexibility. The WT MD simulation model included residues Q14 to S392. The unresolved region between TM12 and TM13 (T334, L335, T336, V337) was modeled using Modeller[56]. The transmembrane domain of SLC26Dg was embedded into 241 POPC lipids and 13185 TIP3P water molecules[57] were added (total system size 77650 atoms). We used GROMACS 5.1.3[58] to perform simulations with a time step of 2 fs at a constant temperature (303.15 K) set with a Nosé-Hoover thermostat[59] using a coupling constant of 1.0 ps. A semi-isotropic Parrinello-Rahman barostat[60] was used to maintain a pressure of 1 bar. The all-atom CHARMM36 force-field was used for the simulation of protein and lipids[61,62]. We performed the monomer MD simulation for ~1.1 μs.

**Modeling of the SLC26Dg dimer using PELDOR time traces**. The conformation of monomeric SLC26Dg at 440 ns of the MD simulation was used for investigation of the dimeric state. We performed rigid-body docking by placing a second protomer against the first protomer. We imposed C2 symmetry by rotating the second protomer in steps of 10 and 5 degrees about axes normal to the membrane plane centered at protomer two and one, respectively, and then bringing the two protomers to contact. For conformations without steric clashes, we forward calculated the PELDOR signals assuming uniform spin-label rotamer distributions[63]. We found that the interface had to be formed by TM13 and TM14 to match the PELDOR data for L385R1. Details on the selection of conformations and dimer modeling procedures are indicated in Supplementary Fig. 5.

**Modeling of the SLC26Dg dimer using distance distributions**. In addition to the docking using PELDOR time traces, the SLC26Dg conformation at 440 ns of the MD simulation was used for rigid-body docking using mean ± SD of the PELDOR distance distributions ($P(r)$s) and C2 symmetry as the restraints. To determine the initial dimer structure model, rigid-body docking was performed using a grid search approach as implemented in the MMMDock tool of the Matlab-based software MMM[55,63]. The experimental distance distributions for the positions

K353R1 (4.4 ± 0.2 nm), V367R1 (3.9 ± 0.3 nm), and L385R1 (1.8 ± 0.1 nm) were used as the restraints, because the corresponding PELDOR time traces show clear oscillations. As the STAS domain of the SLC26Dg monomer is in a non-physiological orientation in the crystal structure, the STAS domain was deleted and the MD refined monomer was used as the starting structure. Assuming C2 symmetry for the dimer ($y = 0$) and a parallel orientation ($z = 0$), a grid of the angle values for $\alpha$ (between 0–360° with 10° steps) and $\beta$ (between 0–180° with 5° steps) and of the translation parameters $x$ and $y$ (between ± 7.5 nm with 0.25 nm steps) was generated.

For each model corresponding to a particular parameter set ($\alpha_i$, $\beta_i$, $x_i$, $y_i$), mean distances for the positions K353R1, V367R1, and L385R1 were simulated. To obtain the initial grid search dimer model, the model with the minimum root mean square deviation (RMSD) to the input values was chosen. The parameter set from this initial search ($\alpha_0 = 360°$, $\beta_0 = 5°$, $x_0 = 5$ nm, $y_0 = 1$ nm) served as the starting point for subsequent refinement, where small changes of the parameters further minimized the RMSD. As the refinement does not sample the whole possible parameter space, it was used after a global grid search to avoid becoming caught in local minima of the error surface. The parameters for the final model are $\alpha = 359.93°$, $\beta = 5.49°$, $x = 5.034$ nm, $y = 1.022$ nm.

**MD simulation of the SLC26Dg dimer**. To relax the SLC26Dg dimer conformation (docked based on PELDOR time traces), we performed an additional MD simulation. The dimer was embedded into 648 POPC lipids and 37227 TIP3P water molecules (total system size 210,115 atoms). We used the same settings for the dimer MD simulation as described above for the monomer MD simulation. We performed the MD simulation for ~ 200 ns.

**BioEn spin-label reweighting**. We calculated PELDOR signals for SLC26Dg dimer conformations saved along the MD simulation at 1-ns intervals. For each saved conformation, we performed a spin-label rotamer refinement[35] by (1) attaching MTSSL labels[63], (2) calculating PELDOR traces for each label position (K353, V367, L385) and rotamer combination, and (3) ensemble-reweighting the spin-label rotamers using the Bayesian inference of ensembles (BioEn)[64,65] maximum-entropy method for each individual dimer conformation. We selected the conformation at 107 ns as the SLC26Dg dimer conformation with minimal total $\chi^2$ for further analysis. L-curve analysis was used to identify suitable confidence parameters $\theta_i$ that trade off consistency between the simulated data at each site and experiment (using a chi-squared metric $\chi^2$) and the changes in the ensemble weights (using relative entropy $S_{KL}^{(i)}$ for label positions $i = 1,2,3$)[35]. Corresponding marginalized reweighted rotamer weights are visualized in Supplementary Fig. 4.

**Radioisotope transport assays**. For transport studies, proteoliposomes were thawed and extruded through a 400 nm polycarbonate filter. Extruded proteoliposomes were pelleted by centrifugation for 20 min at 250,000 × g at 15 °C and resuspended to a final lipid concentration of 100 mg/ml in 50 mM NaPi, pH 7.5, 2 mM MgSO4. The sample was homogenized with a 26 gauge needle and stored at room temperature until use. Radioisotope transport studies were performed on stirred samples at 30 °C. To initiate transport, proteoliposomes were diluted 40-fold into the external buffer (50 mM KPi, pH 6.0, 2 mM MgSO4 containing 24 μM of [14C]-fumarate (Moravek) and 100 nM valinomycin). At appropriate time points, 100 μL samples were taken and immediately diluted with 2 mL ice-cold external buffer, followed by rapid filtration on 0.45 μm nitrocellulose filters. After washing the filters with another 2 mL buffer, the radioactivity associated with the filter was determined by scintillation counting.

**Oxidative cross-linking of SLC26Dg-IL**. SLC26Dg-IL (I45C/A142C, inward-locked) was expressed and purified as described method for cysteine variants used in PELDOR studies. The eluted and HRV 3 C protease-cleaved protein was subjected to SEC in buffer C supplemented with 3 mM DTT. Proteins from peak fractions were pooled and DTT was removed with an Econo-Pac 10DG desalting column (Bio-rad) which was pre-equilibrated with buffer C. The concentration of protein was adjusted to ~ 7 μM and oxidative cross-linking was initiated by adding a 10-fold concentrated CuPhen stock (3 mM CuSO4, 9 mM 1,10-phenanthroline monohydrate, freshly prepared) into the protein solution. The sample was incubated at room temperature for 45 min with gentle agitation and subsequently 0.5 M Na-EDTA, pH 7.0 was added to a final concentration of 20 mM to quench the reaction. The locked protein was injected for SEC to remove the cross-linking reagent and peak fractions were pooled and used for subsequent reconstitution and transport studies.

**Oxidative cross-linking of cysteine mutants along TM14**. *E. coli* MC1061 containing pBXC3sfGH holding the gene coding for cysteine variants of SLC26Dg fused C-terminally to superfolder GFP[66] were cultivated in 700 μL of TB/Amp in a 96 deep well plate. Cells were grown at 37 °C until an OD600 ≈ 1 was reached, after which the temperature was gradually decreased to 25 °C over the course of 1 h. Expression was induced by the addition of 0.005% L-arabinose and proceeded for 16 h. Cells were pelleted and resuspended in 500 μL of 50 mM KPi, pH 7.5, 1 mM MgSO4, 10% glycerol, 3 mM DTT, 1 mM phenylmethylsulfonyl fluoride with trace amounts of DNase I. After 20 min incubation on ice, cell disruption was carried out

by 1 min sonication on ice at output level 4, and a 50% duty cycle (Sonifier 250, Branson). Unbroken cells and debris were removed by 5 min centrifugation at 13,000 × g. The supernatant was collected and DTT was removed by a Bio-Spin 6 column (Bio-rad) which was pre-equilibrated with 50 mM NaPi, pH 7.2. Oxidative cross-linking was initiated by adding a 10-fold concentrated CuPhen stock (3 mM CuSO4, 9 mM 1,10-Phenanthroline monohydrate, freshly prepared) into the vesicle solution. Samples were incubated at room temperature for 20 min and terminated by adding 100 mM N-ethylmaleimide and 0.5 M Na-EDTA, pH 7.0 in a final concentration of 5 mM and 20 mM, respectively. The reaction mixture was mixed with sodium dodecyl sulfate polyacrylamide gel electrophoresis (SDS-PAGE) sample buffer and the degree of cross-linking was determined by 8% SDS-PAGE and in gel GFP fluorescence imaging (ImageQuant LAS4000).

**Preparation of CHO cell membrane vesicles**. CHO/dhFr⁻ cells (ATCC CRL-9096) were cultivated as described previously[67]. After 24 to 36 h of transfection, CHO cells expressing eGFP-tagged rat prestin (cysteine-free) or single-cysteine variants (V499C and I500C located at the cytoplasmic end of TM14[68]) were treated with trypsin, harvested and washed with phosphate-buffered saline (PBS) containing 10 mM DTT. The cells were resuspended in disruption buffer (20 mM HEPES, pH 7.5, 150 mM NaCl, 10% glycerol, 1 mM MgSO4, 3 mM DTT, 20 μg/mL DNase I, 1 tablet of EDTA-free protease inhibitor cocktail) and lysed by six series of 5 sec sonication at 100% amplitude with 1 min incubation on ice in between (Sonoplus GM mini 20, Bandelin). The unbroken cells were removed by centrifugation at 1500 × g for 5 min and the membranes were obtained by ultra-centrifugation at 50,000 rpm for 1 h (TLA110 rotor). The membrane pellet was resuspended in membrane resuspension buffer (20 mM HEPES, pH 7.5, 150 mM NaCl, 10% glycerol, 3 mM DTT) to a final protein concentration of ~ 0.7 mg/mL.

**Oxidative cross-linking in CHO cell membrane vesicles**. The resuspended rat prestin membrane vesicles were pelleted at 200,000 × g for 30 min at 4 °C. After centrifugation, the membrane vesicles were gently washed with 20 mM HEPES, pH 7.5, 150 mM NaCl, 10% glycerol without disturbing the pellet. The cross-linking reaction was initiated by homogeneously resuspension of membrane vesicles in the same amount of cross-linking buffer (20 mM HEPES, pH 7.5, 150 mM NaCl, 10% glycerol, 300 μM CuSO4, 900 μM 1,10-phenanthroline monohydrate, freshly prepared) and incubated at room temperature for 20 min. The cross-linking reaction was quenched by addition of 0.5 M Na-EDTA to a final concentration of 20 mM. To decrease anomalous migration of rat prestin in SDS-PAGE, cross-linked vesicles were treated with PNGaseF (New England BioLabs) at 37 °C for 1.5 h. The reaction mixture was mixed with non-reducing SDS-PAGE sample buffer and the degree of cross-linking was determined by 8% SDS-PAGE and in gel GFP fluorescence imaging (ImageQuant LAS4000).

**Reporting summary**. Further information on research design is available in the Nature Research Reporting Summary linked to this article.

## Data availability

Data supporting the findings of this manuscript are available from the corresponding authors upon reasonable request. A reporting summary for this Article is available as a Supplementary Information file. The source data underlying Figs. 4a, 5b–d, and Supplementary Fig. 2c are provided as a Source Data file. Our structural model of the SLC26Dg dimer is deposited at PDB-Dev under accession code PDBDEV_00000031.

## Code availability

The code used for spin-label rotamer reweighting is freely available at https://github.com/bio-phys/BioEn.

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

## Acknowledgements

We acknowledge professor Thomas Prisner for his support on the EPR measurements. We thank Melanie Engelin and Kai Steinmetz for their support in mutagenesis, and Dr. Brinda Vallat for assistance with the PDB-Dev deposition. We acknowledge financial support from the Max Planck Society (KR, GH), the International Max Planck Research School (IMPRS) of the MPI of Biophysics in Frankfurt (EAJ), and the German Research Foundation via the Cluster of Excellence Frankfurt (Macromolecular Complexes; GH, ERG), the SPP1608 (Ultrafast and temporally precise information processing: normal and dysfunctional hearing; OL 240/4–2 to DO), and the SFB807 (Transport and Communication across Biological Membranes; GH, BJ, ERG).

## Author contributions

YNC prepared samples for EPR and designed and performed cross-linking and functional studies under supervision of ERG. EAJ designed and performed EPR measurements under supervision of BJ. KR designed and performed MD simulations and spin-label ensemble refinement under supervision of GH. JH expressed rat prestin under supervision of DO. GH, BJ, and ERG designed the research. ERG drafted the manuscript. All authors analyzed and interpreted the data and contributed to the manuscript.

## Additional information

**Competing interests:** The authors declare no competing interests.

