## [Peer Review File · Nature Communications]

Reviewers' Comments:

Reviewer #1:

Remarks to the Author:

Summary:

The authors present research on an SLC26 protein from *Deinococcus geothermalis*, concentrated on mapping the predicted dimerization interface and investigating its importance for protein function. The authors use the available crystal structure of their protein of interest in its monomeric form and, through distance measurements from DEER EPR and subsequent MD experiments, model a dimer of SLC26Dg. Through the use of DEER and a deletion mutant of SLC26Dg (lacking its cytoplasmic domain STAS), they show that STAS is not required for dimerization but might play a stabilizing role in this process. Another piece of evidence in support of the dimer model comes from oxidative cross-linking of single cysteine mutants on TM14, done in *E. coli* cells. This experiment was also replicated in two other organisms, and led authors to believe that the dimerization interface of SLC26 family members is conserved. A fumarate transport assay in liposomes served as a tool to evaluate functional relevance of SLC26Dg dimer, where activity of heterodimers with inactive SLC26Dg protein variant was measured. The paper expanded the existing understanding of structural and functional role of the dimer interface in SLC26 family of proteins.

Comments:

- Results, page 7, paragraph 1: Figure 4B could be used to look at the results, so mark it in text (Fig 4B). Also, in that figure, remove zoomed-in view of the helix as it is redundant (colors on the figure of the whole monomer indicate experimental results just fine), (suggest -removing the lighter-darker coloring type and adding a panel showing interactions of TM14 in a dimer with cross-linking results mapped).
- Results, page 7, paragraph 3: "...reconstituted (Fig. 5c)." – lines on the figure have very similar colors, very hard to read. Also, in the legend, mention the number of experiment repeats.
- Results, page 8, paragraph 1: Please describe how you derived the "anticipated curves assuming an activity of the heterodimers corresponding to 0, 50, and 100% of the wildtype homodimers". Also, in the Figure 5d, as for Figure 5c, please change the colors of the curves. In addition, the figure would improve with better placed labels: the legend for the curves (100, 50 and 0%) should indicate that those are your models with expected activity of heterodimers; the legend for the X axis should indicate that this is the mixing ratios of wild type and crosslinked/mutant SLC26Dg.
- Discussion, page 9, paragraph 1: "Further cross-linking studies on additional prokaryotic and mammalian homologues suggest that this interface is evolutionary conserved in the SLC26 family." – This is a bold statement. Perhaps saying that "it might be" conserved is more appropriate. A justification for drawing this conclusion after investigating just one mammalian protein would need to be included? You could make a figure with sequence alignment and % sequence identity of a number of organisms (include humans and reference Dg), and then discuss the alignments around the regions of dimerization.

Minor Comments

- Both DEER and PELDOR are abbreviations that can be used to describe the technique (correct?) thus both should be mentioned in the abstract. This should also allow for better searching for your paper.
- In the discussion section, authors might want to mention techniques for getting a structure of SLC26Dg: in addition to X-ray and troubleshooting crystallization through things like removal of STAS domain or crystallization of fusion proteins, you might want to add cryoEM (your GFP-fusion protein should give you a C2 symmetry dimer at ~120 kDa which could be targeted with this technique in the future).
- In the discussion section, authors might want to discuss the possibility of only 1 protomer being functionally active in the liposome assays, which would explain the high activity of heterodimers.

This is mentioned in the results and then abandon it in the discussion, and that is an important point to talk about.

- Introduction, page 3, paragraph 2: "carboxyl-terminal" – should be "carboxy-terminal"
- Results, page 6, paragraph 1: "...shown in Fig. 3" – indicate TM13 and 14 on the figure
- Results, page 6, paragraph 2: "...interfaces of SLC4 and SLC23 proteins center around..." – indicate these helices on Figure 1
- Results, page 6, paragraph 2: "...the midpoint of the SLC26Dg dimer is TM14 on the opposite side of the gate domain." – This is unclear, what does "opposite side of the gate domain" mean here? It is enough to say that TM14 are the dimer interface in SLC26Dg.
- Results, page 6, paragraph 2: "Furthermore, while the membrane dimer interface of SLC4 and SLC23 proteins involves extensive interactions covering approximately half and nearly the complete exposed membrane surface of their gate domains, respectively, the membrane interface of SLC26Dg is relatively small." This reads poorly. Work on it. Changing "and nearly the complete" to "or almost all" could help.
- Results, page 7, paragraph 1: mention in the text that this protein is a GFP fusion
- Results, page 7, paragraph 1: "...a band with lower electrophoretic mobility (Fig 4) " – add Fig 4A to be more specific
- Results, page 7, paragraph 2: "...norvegicus." – give % sequence identity between your two newly tested proteins and the reference SLC26Dg.
- Results, page 7, paragraph 2: for Fig 5A, please label helices referenced in the text (TM1 and TM5).
- Discussion, page 9, paragraph 2: "The structures of dimeric SLC4 and SLC23 proteins in different conformations all hold identical, yet family-specific, contact surfaces". This sentence is not built correctly as it implies that there is just 1 contact surface for all families and adding 'yet family-specific' does not do much to change how this sentence reads. Change to something like this: "The structures of dimeric SLC4 and SLC23 proteins in different conformations have identical contact surfaces within each family"
- Discussion, page 9, paragraph 2: "...cytoplasmic region following TM12 in SLC4A1 and SLC4A4..." – Shouldn't there be references here?
- Discussion, page 9, paragraph 3: "...stabilize an SLC23 dimer and other oligomeric membrane proteins 36, 43." – Put the reference to SLC23 directly after "SLC23 dimer", to distinguish it from the other reference.
- Discussion, page 10, paragraph 2: "...inferred in diverse ways...". Should be "interfered"
- Discussion, page 10, paragraph 2: move reference number 29 to the end of the last sentence in this paragraph.

Reviewer #2:

Remarks to the Author:

The manuscript "Structural basis for functional interactions in dimers of SLC26 transporters" by Chang et al. presents an interesting account combining EPR spectroscopy, molecular dynamics simulations and biochemical analysis to infer the structure of the functional SLC26 dimer and investigate its functional relevance. SLC26 (and other 7TMIR proteins) are important secondary transporters that maintain anion equilibria and have implications in several diseases.

This study is providing a model for the *Deinococcus geothermalis* SLC26 protein dimer where monomeric SLC26 protein structures are established in literature but structures of oligomeric functional complexes had been elusive. SLC26 is consistent with 7TMIR proteins forming structural dimers in the membrane, however the dimerisation interface is significantly smaller than in other 7TMIR protein dimers.

In addition to this specific output the study also nicely showcases the power of combining sparse EPR distance data with functional analyses and biochemical methods. As such it should provide an outstanding example to other researchers in the fields and inspire progress in unravelling the

structures and their-functional implications for numerous elusive membrane protein complexes. The relatively small amount of constraints and the fact that emerging methodology is showcased to a broader audience both mandate that the data is scrutinized as detailed as possible. In this regard the greatest weakness of the manuscript is the incomplete presentation, analysis and contextualization of the EPR data. Once these have been rectified the manuscript should be publishable in Nature Communications.

Major

DEER distances:

The narrow distance distribution for three mutants in fig 2 are beyond doubt. Some might miss the raw data but that finally appears in fig S5 (I would have preferred the full data in fig S2). These 3 constructs fully support the model and the conclusion with the grain of salt that the data is extremely limited and there might be further structures satisfying the constraints.

The significance of a distance obtained depends on the length of the form factor as the authors rightly state. However, the authors fail to discuss the reliability of their distance analysis which they also overstretch in fig S2. It would be very helpful if the authors added a comprehensive statement to this to the supplementary and refer to this in the manuscript. This should cover the points made in the following.

The data in fig S2 is overinterpreted. Assuming the authors did not magically circumvent the distance limitations of DEER spectroscopy, formidably explained in the DeerAnalysis manual (epr.ethz.ch) the interpretation of data in fig S2 is not valid. The shapes of the $P(r)$ in fig S2A are irrelevant as they are not reliable only the mean is significant above 5 nm for these form factor lengths and there might be some deviation the V129R1 position with respect to the model.

The authors further state that the distances in the mutants in panel B are expected >10 nm and will "already" (very euphemistic) be lost in the background. For form factors of 2 and 4 us everything above 6 and 7.5 nm, respectively, will be lost in the background.

All distances discussed in the caption of supplement fig 2 for panel C are in a region that cannot be quantified even for the longest form factor the authors present (for form factors of 2 and 4 us everything above 5 and 6.3 nm, respectively indicates presence of a long distance but cannot be attributed a number). Thus, the discussion around the five positions "indirectly" validating the model is neither backed by data or scientifically sound. If it were it should have been compared to the model. (This would anyways have been meaningless as the data for these mutants in not quantifiable.)

Phase memory times:

The authors find very quick dephasing. Have higher proportions of phospholipids been tried to make sure this is not dominated by local concentration. How was TM determined? Why is the raw data not shown?

Spin-labelling efficiencies:

Why are the cw EPR data not shown, why are the individual labelling degrees not given? Do the individual labelling degrees and DEER modulation depths correspond?

Dimer formation in detergent/phospholipid membrane:

Has the protein concentration in/with respect to detergent been increased to check for dimers with high K_d ? Has the protein been diluted with respect to phospholipid to see if the modulation depth changes. This could give limits for the K_d in both environments.

Deletion of STAS:

The interpretation of an increased flexibility and distance is not well founded. The inter dimer distance between the L385R1 pair is very narrowly distributed, probably by steric constraints, the distribution with changes with deletion of STAT so that the sterics around the labels or the geometry between the labelling sites must have changed. For the other two constructs, there is a

complete loss of oscillation in the form factor meaning there is no evidence for the regions containing the labels to be structured.

Minor

The introduction promises “the first structural and mechanistic insights in the allosteric interactions between SLC26 protomers.” However, this is never explicitly taken up in the results and discussion.

The authors produced 13 folded and active spin-labelled mutants of SLC26Dg (p5). However, fig 1 and 2+S2 suggest 14 mutants.

The authors do not state how P(r) were generated, but if this was Tikhonov regularisation in DeerAnalysis this should be mentioned and referenced.

Reviewer #3:

Remarks to the Author:

The authors characterize the dimeric structure of an SLC26 transporter from *D. geothermalis*, using a range of biophysical, biochemical, and computational techniques. The results suggest a novel mode of dimerization of the SLC26 family, differing from those observed in other related families such as SLC4 and SLC23.

Overall this is a well-written and very interesting manuscript that provides some novel findings regarding the mechanism and evolution of the SLC26 family and other elevator transporters. Not too much is known about the structure and mechanism of the SLC26 family and other related proteins, so this study is timely.

I have some comments that are mostly related to the computational methods and the interpretation of their outputs, as well as suggestions related to the manuscript's organization which I think would improve its readability.

1) To predict the dimer interface, the authors use protein-protein docking with some experimentally derived constraints. Protein-protein docking can be inaccurate. Were there any other solutions proposed by docking? Did they converge? Alternative solutions predicted by the programs should be provided and refuted. Furthermore, a discussion about the uncertainty of the calculations should be added.

2) If I understand correctly, docking was done on one conformation from the MD trajectories (a frame from 440 ns). Again, due to the limitation in docking, it would be important to see whether different solutions are predicted when different trajectories (or models) are docked. For example, one suggestion would be to cluster the entire trajectory, and then dock 3-5 relevant conformations and show some of the docking solutions in the supplementary material.

3) The dimeric model suggests that the interface between the protomers is uniquely small (350 \AA^2). I could imagine that a small change in the composition of the interface may have an effect on the dimer. Did the authors test whether some mutations break the complex formation (computationally and/or experimentally)? This would substantiate the prediction significantly.

4) It will make it much easier to understand the manuscript, if Figure S4C-3 (with the residue visualization) will be integrated into Figure 2, and Figure S10E into Figure 5. In addition, the more detailed description of the docking protocol in the supplementary material should be a part of the

main text.

Reviewers' comments:

Reviewer #1 (Remarks to the Author):

Summary:

The authors present research on an SLC26 protein from *Deinococcus geothermalis*, concentrated on mapping the predicted dimerization interface and investigating its importance for protein function. The authors use the available crystal structure of their protein of interest in its monomeric form and, through distance measurements from DEER EPR and subsequent MD experiments, model a dimer of SLC26Dg. Through the use of DEER and a deletion mutant of SLC26Dg (lacking its cytoplasmic domain STAS), they show that STAS is not required for dimerization but might play a stabilizing role in this process. Another piece of evidence in support of the dimer model comes from oxidative cross-linking of single cysteine mutants on TM14, done in *E. coli* cells. This experiment was also replicated in two other organisms, and led authors to believe that the dimerization interface of SLC26 family members is conserved. A fumarate transport assay in liposomes served as a tool to evaluate functional relevance of SLC26Dg dimer, where activity of heterodimers with inactive SLC26Dg protein variant was measured. The paper expanded the existing understanding of structural and functional role of the dimer interface in SLC26 family of proteins.

We thank the Reviewer for the constructive comments and suggestions to improve our manuscript.

Comments:

- Results, page 7, paragraph 1: Figure 4B could be used to look at the results, so mark it in text (Fig 4B). Also, in that figure, remove zoomed-in view of the helix as it is redundant (colors on the figure of the whole monomer indicate experimental results just fine), (suggest -removing the lighter-darker coloring type and adding a panel showing interactions of TM14 in a dimer with cross-linking results mapped).

We have revised Fig. 4 to replace the redundant panel with a new panel (Fig. 4b) showing the position of TM14 in the context of the dimer and with the cross-linking results mapped on its surface. We revised the text to refer to Fig. 4b.

- Results, page 7, paragraph 3: "...reconstituted (Fig. 5c)." – lines on the figure have very similar colors, very hard to read. Also, in the legend, mention the number of experiment repeats

We have revised the line colors in Fig 5c and 5d as suggested. We have revised the legend to indicate that all data points represent the mean of three replicates.

- Results, page 8, paragraph 1: Please describe how you derived the "anticipated curves assuming an activity of the heterodimers corresponding to 0, 50, and 100% of the wildtype homodimers". Also, in the Figure 5d, as for Figure 5c, please change the colors of the curves. In addition, the figure would improve with better placed labels: the legend for the curves (100, 50 and 0%) should indicate that those are your models with expected activity of heterodimers; the legend for the X axis should indicate that this is the mixing ratios of wild type and crosslinked/mutant SLC26Dg.

We have revised the legend of Fig. 5 to indicate how the modelled curves were constructed.

"These models were calculated assuming stochastic dimer formation (e.g., mixing WT:IL protomers in a 50:50 ratio results in 25% WT-WT, 50% WT-IL, and 25% IL-IL dimers) and specific transport activities of 32.3 or 6.8 nmol fumarate per mg WT or IL homodimer per min, respectively, and heterodimer activities corresponding to 0, 50, or 100% of WT homodimer."

We have revised the line colors in Fig 5c and 5d as suggested.

We have revised the legend of Fig. 5d to specify that the curves represent models as suggested. To illustrate this additionally, we have used discontinuous lines.

Furthermore, we have specified in the X-axis label that the percentages indicate mixing ratios.

- Discussion, page 9, paragraph 1: "Further cross-linking studies on additional prokaryotic and mammalian homologues suggest that this interface is evolutionary conserved in the SLC26 family." – This is a bold statement. Perhaps saying that "it might be" conserved is more appropriate.

We agree with the Reviewer and revised the statement as suggested.

A justification for drawing this conclusion after investigating just one mammalian protein would need to be included? You could make a figure with sequence alignment and % sequence identity of a number of organisms (include humans and reference Dg), and then discuss the alignments around the regions of dimerization.

Our suggestion that the specific interface identified in our study might be evolutionarily conserved in SLC26 proteins originates from our appreciation of structurally conserved dimer interfaces in the SLC4 and SLC23 families, which in the case of the SLC23 family has been demonstrated to be conserved across kingdoms.

We have performed the suggested sequence alignment of TM14 (Suppl. Fig. 13) for 11 mammalian and 11 prokaryotic SLC26 proteins. This alignment does not indicate a strong sequence conservation between mammalian and prokaryotic SLC26 proteins. We do note a GxxxG-like motif towards the extracellular side of TM14. Though these motifs often mediate dimerization in single-pass membrane proteins, in multi-pass membrane proteins they seem more likely to be involved in protein folding. Furthermore, this specific region of TM14 is not directly involved in protomer-protomer interactions in our model. Specificity of the SLC26 protomer-protomer interaction may instead arise from general complementarity of the interacting surfaces combined with other, potentially conserved features such as interfacial lipids. As we currently do not know the identity of these factors, nor where they could bridge the protomers, we agree with the Reviewer that we have to phrase our suspicion of interface conservation carefully.

We have added this additional discussion to the first paragraph in the Discussion section where we mention the potential conservation of the interface.

Minor Comments

- Both DEER and PELDOR are abbreviations that can be used to describe the technique (correct?) thus both should be mentioned in the abstract. This should also allow for better searching for your paper.

This is correct. We thank the Reviewer for this suggestion. We have adjusted the respective sentence to: "...and characterize its functional relevance by combining PELDOR/DEER distance measurements..."

- In the discussion section, authors might want to mention techniques for getting a structure of SLC26Dg: in addition to X-ray and troubleshooting crystallization through things like removal of STAS domain or crystallization of fusion proteins, you might want to add cryoEM (your GFP-fusion protein should give you a C2 symmetry dimer at ~120 kDa which could be targeted with this technique in the future).

We prefer to keep the discussion focused to get our main message on the interpretation of our structural and functional data across. We thus like to refrain from expanding on potential methodological strategies for obtaining a structure of an SLC26 dimer directly.

Shortly: in the past years, we have extensively tried to obtain an X-ray structure of an SLC26 dimer, including removal of the STAS domain (which regrettably resulted in the structure of a monomeric truncation mutant; mentioned in Geertsma et al. 2015) and use of crystallization chaperones. We agree with the Reviewer that cryoEM offers exciting new opportunities for obtaining the structure of a dimeric SLC26. Regrettably, the GFP-fusion is in this respect not particularly helpful as this does not stably interact with the protein and remains highly flexible, thereby complicating the data analysis.

- In the discussion section, authors might want to discuss the possibility of only 1 protomer being functionally active in the liposome assays, which would explain the high activity of heterodimers. This

is mentioned in the results and then abandon it in the discussion, and that is an important point to talk about.

We agree with the Reviewer that this option should also be mentioned in the discussion section. We have revised the text to:

“Though the similar transport rates of wildtype homodimers and WT-IL heterodimers may also imply that only one protomer is active in the SLC26Dg dimer, the latter appears in conflict with the intermediate non-linear capacitance observed for rat prestin heterodimers [Detro-Dassen et al. 2008].”

- Introduction, page 3, paragraph 2: “carboxyl-terminal” – should be “carboxy-terminal”

We replaced “carboxyl-terminal” with “carboxy-terminal” throughout the manuscript.

- Results, page 6, paragraph 1: “...shown in Fig. 3” – indicate TM13 and 14 on the figure

We appropriately labeled the transmembrane segments of the gate domain in Fig. 3a.

- Results, page 6, paragraph 2: “...interfaces of SLC4 and SLC23 proteins center around...” – indicate these helices on Figure 1

We labeled the respective TMs in Fig. 1a and 1b.

- Results, page 6, paragraph 2: “...the midpoint of the SLC26Dg dimer is TM14 on the opposite side of the gate domain.” – This is unclear, what does “opposite side of the gate domain” mean here? It is enough to say that TM14 are the dimer interface in SLC26Dg.

We used this statement to emphasize how different the SLC26 interface is from those observed for the SLC4 and SLC23 family, that is: the main interactions take place in a very different section of the gate domain. We agree with the Reviewer that this phrasing may lead to confusion and have revised this section as suggested.

“Whereas the membrane dimer interfaces of SLC4 and SLC23 proteins center around TM6, and TM5 plus TM12, respectively, the midpoint of the SLC26Dg dimer is TM14.”

- Results, page 6, paragraph 2: “Furthermore, while the membrane dimer interface of SLC4 and SLC23 proteins involves extensive interactions covering approximately half and nearly the complete exposed membrane surface of their gate domains, respectively, the membrane interface of SLC26Dg is relatively small.” This reads poorly. Work on it. Changing “and nearly the complete” to “or almost all” could help.

We thank the Reviewer for this suggestion. We have revised the sentence to read:

“Furthermore, while the membrane dimer interface of SLC4 and SLC23 proteins involves extensive interactions covering large fractions of the exposed membrane surface of their gate domains, the membrane interface of SLC26Dg is relatively small.”

- Results, page 7, paragraph 1: mention in the text that this protein is a GFP fusion

We added a statement in the respective paragraph to indicate that SLC26Dg-fusions to superfolder-GFP were used.

“Oxidative cross-linking of single-cysteine variants at several positions in TM14 of SLC26Dg, fused to superfolder-GFP to facilitate detection, lead to the appearance of a band with lower electrophoretic mobility (Fig. 4a).”

- Results, page 7, paragraph 1: "...a band with lower electrophoretic mobility (Fig 4)" – add Fig 4A to be more specific

We revised the text to indicate specifically what panel we refer to.

- Results, page 7, paragraph 2: "...norvegicus." – give % sequence identity between your two newly tested proteins and the reference SLC26Dg.

We revised to mention the percentage sequence identity explicitly:

"To test this, we used the same TM14 cross-linking approach on SLC26 proteins from *Sulfitobacter indolifex* and *Rattus norvegicus*, that hold 23% and 21% sequence identity to SLC26Dg, respectively (Suppl. fig 7-8)."

- Results, page 7, paragraph 2: for Fig 5A, please label helices referenced in the text (TM1 and TM5).

Fig. 5a serves to indicate the positions of the cysteine mutations and their relevance in locking the transporter in an inward-facing conformation. Based on the suggestion of Reviewer #3 we have revised Fig. 5a to emphasize the inward-facing conformation by showing the water-filled funnel. This clipping representation obscures the TMs in the figure and consequently, adding labels for the helices would not make the figure clearer. Instead, we have revised the text to place more emphasis on the positions 45 and 142 that are clearly indicated in the figure.

"Based on the crystal structure, we selected Ile-45 on the extracellular side of TM1 (core) and Ala-142 in TM5 (gate) as most suited positions concerning cross-linking efficiency and ability to lock the protein and prevent transport (Fig. 5a)."

- Discussion, page 9, paragraph 2: "The structures of dimeric SLC4 and SLC23 proteins in different conformations all hold identical, yet family-specific, contact surfaces". This sentence is not built correctly as it implies that there is just 1 contact surface for all families and adding 'yet family-specific' does not do much to change how this sentence reads. Change to something like this: "The structures of dimeric SLC4 and SLC23 proteins in different conformations have identical contact surfaces within each family"

We thank the Reviewer for this suggestion. We have revised the text as suggested.

"The structures of dimeric SLC4 and SLC23 proteins in different conformations have identical contact surfaces within each family."

- Discussion, page 9, paragraph 2: "...cytoplasmic region following TM12 in SLC4A1 and SLC4A4..." – Shouldn't there be references here?

We have added the references to the respective papers presenting these structures (Arakawa et al. 2015; Huynh et al., 2018).

- Discussion, page 9, paragraph 3: "...stabilize an SLC23 dimer and other oligomeric membrane proteins 36, 43." – Put the reference to SLC23 directly after "SLC23 dimer", to distinguish it from the other reference.

We have adjusted the position of this reference.

- Discussion, page 10, paragraph 2: "...inferred in diverse ways...". Should be "interfered"

We thank the Reviewer for this correction. Text revised.

- Discussion, page 10, paragraph 2: move reference number 29 to the end of the last sentence in this paragraph.

We have adjusted the position of this reference.

Reviewer #2 (Remarks to the Author):

The manuscript “Structural basis for functional interactions in dimers of SLC26 transporters” by Chang et al. presents an interesting account combining EPR spectroscopy, molecular dynamics simulations and biochemical analysis to infer the structure of the functional SLC26 dimer and investigate its functional relevance. SLC26 (and other 7TMIR proteins) are important secondary transporters that maintain anion equilibria and have implications in several diseases.

This study is providing a model for the *Deinococcus geothermalis* SLC26 protein dimer where monomeric SLC26 protein structures are established in literature but structures of oligomeric functional complexes had been elusive. SLC26 is consistent with 7TMIR proteins forming structural dimers in the membrane, however the dimerisation interface is significantly smaller than in other 7TMIR protein dimers.

In addition to this specific output the study also nicely showcases the power of combining sparse EPR distance data with functional analyses and biochemical methods. As such it should provide an outstanding example to other researchers in the fields and inspire progress in unravelling the structures and their-functional implications for numerous elusive membrane protein complexes. The relatively small amount of constraints and the fact that emerging methodology is showcased to a broader audience both mandate that the data is scrutinized as detailed as possible. In this regard the greatest weakness of the manuscript is the incomplete presentation, analysis and contextualization of the EPR data. Once these have been rectified the manuscript should be publishable in Nature Communications.

We thank the Reviewer for the positive comments and suggestions to improve our manuscript.

Major

DEER distances:

The narrow distance distribution for three mutants in fig 2 are beyond doubt. Some might miss the raw data but that finally appears in fig S5 (I would have preferred the full data in fig S2). These 3 constructs fully support the model and the conclusion with the grain of salt that the data is extremely limited and there might be further structures satisfying the constraints.

The significance of a distance obtained depends on the length of the form factor as the authors rightly state. However, the authors fail to discuss the reliability of their distance analysis which they also overstretch in fig S2. It would be very helpful if the authors added a comprehensive statement to this to the supplementary and refer to this in the manuscript. This should cover the points made in the following.

The data in fig S2 is overinterpreted. Assuming the authors did not magically circumvent the distance limitations of DEER spectroscopy, formidably explained in the DeerAnalysis manual (epr.ethz.ch) the interpretation of data in fig S2 is not valid. The shapes of the $P(r)$ in fig S2A are irrelevant as they are not reliable only the mean is significant above 5 nm for these form factor lengths and there might be some deviation the V129R1 position with respect to the model.

Following this constructive criticism, we made further changes in Suppl. Fig. 4 (previously Fig. S2). For the data in panel A, we now present the $P(r)$ data with additional color coding to indicate the reliability of the mean, width, and shape of the probability distribution. As the Reviewer pointed out, now it clearly shows that only the mean is reliable for distances >5 nm.

In addition, we have revised the text to soften our statement that the additional EPR data validate our dimer model. Instead, we now indicate that the data are in agreement with the dimer model.

“ Simulations performed on this model for two additional positions, V129R1 (TM5) and L248R1 (TM8), agree with the experimental data as well (Suppl. Fig. 4a). For the other nine positions in the gate domain, simulations predict a mean interspin distance in the range of 6.3-10.6 nm, which could not be accurately determined due to the short T_M (Suppl. Fig. 4b and Suppl. Fig 3).”

The authors further state that the distances in the mutants in panel B are expected >10 nm and will “already” (very euphemistic) be lost in the background. For form factors of 2 and 4 us everything above 6 and 7.5 nm, respectively, will be lost in the background.

We have removed this statement and now clearly indicate:

“For the other nine positions in the gate domain, simulations predict a mean interspin distance in the range of 6.3-10.6 nm, which could not be accurately determined due to the short T_M (Suppl. Fig. 4b and Suppl. Fig 3).”

All distances discussed in the caption of supplement fig 2 for panel C are in a region that cannot be quantified even for the longest form factor the authors present (for form factors of 2 and 4 us everything above 5 and 6.3 nm, respectively indicates presence of a long distance but cannot be attributed a number). Thus, the discussion around the five positions “indirectly” validating the model is neither backed by data or scientifically sound. If it were it should have been compared to the model. (This would anyways have been meaningless as the data for these mutants in not quantifiable.)

We agree with the Reviewer that none of these data are long enough to quantify the distance distributions (which was not our aim either) and we should have better described them. We combined the rest of the data into panel B with a statement that simulations on the model predict mean distances between 6.3-10.6 nm for these positions. Due to the limited time window for dipolar evolution none of those distances could be determined. Also, we removed the statement that the five positions (earlier in panel C) indirectly validate the model. Now we conclude that “although the data indicate the presence of long distances as the simulations predict, owing to short T_M the data could not be acquired long enough to determine those distances” (legend Suppl. Fig. 4).

Phase memory times:

The authors find very quick dephasing. Have higher proportions of phospholipids been tried to make sure this is not dominated by local concentration.

How was T_M determined?

Why is the raw data not shown?

The reconstitution was done at a 1:20 weight ratio and a 1:1400 molar ratio of protein-to-lipids. Given our protein reconstitution efficiencies of 40-50%, we are already at the limit of the sensitivity for PELDOR experiments and therefore did not try lower protein-to-lipid ratios.

We agree that the T_M values we observed are comparably short. However, our values between 1-2 μ s are in a range commonly observed for MTSSL at buried sites in a protein [Huber et al. 2001]. Furthermore, the low values may be explained by the location of our labels in close proximity to the membrane, which can further reduce T_M . [Borbat et al. 2013]

The T_M values were determined using a 2-pulse echo decay as a function of the interpulse delay. The time at which the signal intensity drops to 1/e is given as the T_M .

We apologize for not showing the raw data. In the revised manuscript, we present all the original data in Suppl. Fig. 3.

*Huber, M.; Lindgren, M.; Hammarstrom, P.; Martensson, L.-G.; Carlsson, U.; Eaton, G. R.; Eaton, S. S. Phase Memory Relaxation Times of Spin Labels in Human Carbonic Anhydrase II: Pulsed EPR to Determine Spin Label Location. *Biophys. Chem.* 2001, 94, 245–256.*

*Borbat, P. P., Georgieva, E. R. & Freed, J. H. Improved Sensitivity for Long-Distance Measurements in Biomolecules: Five-Pulse Double Electron-Electron Resonance. *J. Phys. Chem. Lett.* 4, 170-175 (2013).*

Spin-labelling efficiencies:

Why are the cw EPR data not shown, why are the individual labelling degrees not given? Do the individual labelling degrees and DEER modulation depths correspond?

We apologize for this inconvenience. In our revised manuscript we now show the cw EPR data for all the positions in both detergent micelles and in proteoliposomes, including the comparison between full-length and STAS domain truncation variant, in Suppl. Fig. 1. The labelling efficiency for the different positions is indicated in Suppl. Table 1. These values varied between 70-100%.

The observed modulation depth of positions for which reliable data could be measured agrees with the dimerization. These values are also indicated in Suppl. Table 1. Position 353 seems to be an outlier with a somewhat lower value, which may be due to either an overestimation of the labeling efficiency or the loss of spin labels during reconstitution.

Dimer formation in detergent/phospholipid membrane:

Has the protein concentration in/with respect to detergent been increased to check for dimers with high Kd? Has the protein been diluted with respect to phospholipid to see if the modulation depth changes. This could give limits for the Kd in both environments.

We agree with the Reviewer that the Kd in detergent and lipid membranes would be interesting to know, though this would be outside the scope of this study. Furthermore, practical and biological reasons prevent us from following the suggested experimental path. As discussed above, our measurements on membrane-reconstituted SLC26Dg are already at the limit of the sensitivity which prevents us from using lower protein concentrations. In detergent, we observe predominantly monomers of SLC26Dg, irrespective of detergent type and protein concentration. Even in our protein crystals, where protein concentration is very high, the protein remains monomeric. We speculate that extrinsic factors such as interfacial lipids are required to stabilize the SLC26 dimer interface.

Deletion of STAS:

The interpretation of an increased flexibility and distance is not well founded. The inter dimer distance between the L385R1 pair is very narrowly distributed, probably by steric constraints, the distribution width changes with deletion of STAS so that the sterics around the labels or the geometry between the labelling sites must have changed. For the other two constructs, there is a complete loss of oscillation in the form factor meaning there is no evidence for the regions containing the labels to be structured.

We agree with the Reviewer that changes at position L385R1 could be explained by a rotamer rearrangement. As we could not measure K353R1 and V367R1 data long enough, we were unable to determine whether the above interpretation holds true for these positions as well or an increased flexibility accounts for the observed changes in the PELDOR data. We made changes accordingly in the manuscript.

“STAS domain deletion resulted in a small increase in the mean L385R1 distance from 1.8 ± 0.1 to 2.1 ± 0.1 nm, that, given the narrow distance distribution, rather suggests a rearrangement of the MTSSL rotamers than a physical separation of the protomers. The complete disappearance of oscillations in the primary PELDOR data of SLC26Dg^{ΔSTAS}-K353R1 and -V367R1 in TM13 suggests that either similar rearrangements of spin label rotamers or an increased flexibility at these positions may underlie these changes (Suppl. Fig 8). The latter could not be confirmed due to the limited time window of the dipolar evolution. Thus, while deletion of the STAS domain appears to affect the environment around the spin labels in TM13 and TM14, the STAS domain itself is not a prerequisite for dimerization.”

Minor

The introduction promises “the first structural and mechanistic insights in the allosteric interactions between SLC26 protomers.” However, this is never explicitly taken up in the results and discussion.

With “first structural [...] insights in the allosteric interactions between [...] protomers” we refer to the structural model that we have generated. This is extensively discussed throughout the manuscript.

With “first [...] mechanistic insights in the allosteric interactions between [...] protomers” we refer to our interpretation of the functional cooperativity in SLC26Dg dimers. This can be found in the Discussion section starting with “Though the mechanistic basis for functional interactions in 7TMIR dimers is currently unclear, important insights were obtained from the characterization of monomeric 7TMIR proteins.”

In essence, we provide three hypotheses on the role of the dimerization for the transport mechanism in the Discussion section: 1) comparison of the UraA monomer/dimer data [Yu et al. 2017] and our data may be explained by the fact that dimerization leads to a subtle rearrangement of the gate domains, thereby enabling transport; 2) the apparently increased transport rate of the WT-IL heterodimer may highlight the relevance of a large stably-embedded domain to facilitate the conformational change of the mobile transport domain (core domain); 3) or alternatively, the apparently increased transport rate may indicate that the transport mechanism involves only one active protomer.

At the moment, our data does not allow more detailed interpretation, hence we prefer not to expand this section further. We hope this clarification suffices.

The authors produced 13 folded and active spin-labelled mutants of SLC26Dg (p5). However, fig 1 and 2+S2 suggest 14 mutants.

We engineered 13 positions in the gate domain and one additional position in TM8 of the core domain. We have revised the text to clarify this apparent discrepancy.

“One additional central position in the core domain (TM8) was also selected.”

The authors do not state how P(r) were generated, but if this was Tikhonov regularisation in DeerAnalysis this should be mentioned and referenced.

Interspin distances were determined using the DeerAnalysis software employing Tikhonov regularization. In the revised manuscript we now describe the PELDOR data processing in the Materials and Methods section with appropriate references.

“Distance distributions were determined using DeerAnalysis [Jeschke et al. 2006f]. The normalized primary PELDOR data $V(t)/V(0)$ were processed to remove the intermolecular contribution and the resulting form factors $F(t)/F(0)$ were fitted with a model-free Tikhonov regularization to determine the distance distributions. The MATLAB-based MMM [Jeschke, 2018] software was used for simulation of interspin distances on the form factor-based dimer model.”

Reviewer #3 (Remarks to the Author):

The authors characterize the dimeric structure of an SLC26 transporter from *D. geothermalis*, using a range of biophysical, biochemical, and computational techniques. The results suggest a novel mode of dimerization of the SLC26 family, differing from those observed in other related families such as SLC4 and SLC23.

Overall this is a well-written and very interesting manuscript that provides some novel findings regarding the mechanism and evolution of the SLC26 family and other elevator transporters. Not too much is known about the structure and mechanism of the SLC26 family and other related proteins, so this study is timely.

I have some comments that are mostly related to the computational methods and the interpretation of their outputs, as well as suggestions related to the manuscript's organization which I think would improve its readability.

We thank the Reviewer for the positive comments and suggestions to improve our manuscript.

1) To predict the dimer interface, the authors use protein-protein docking with some experimentally derived constraints. Protein-protein docking can be inaccurate. Were there any other solutions proposed by docking? Did they converge? Alternative solutions predicted by the programs should be provided and refuted.

We agree with the Reviewer, that in general protein-protein docking approaches can be inaccurate and that unique solutions are often difficult to identify. Here, we take advantage of the fact that binding takes place in the membrane, which greatly reduces the space of possible docking solutions. To identify the interface of members of the SLC26 family, we performed an unbiased rigid-body protein-protein docking of two protomers in the membrane plane and rotated one protomer around the other under a C_2 -symmetry constraint. In this way, the dimers are uniquely defined by the polar angle defining the in-plane rotation of the second protomer. For all SLC26 dimers generated by docking, we then calculated the PELDOR signals for the different label positions. For each label position, we plotted the reduced χ^2 as a function of the polar plane angle, which allowed us to assess the consistency of experimental and simulated data for each conformation. In this way, we found that the interface can only be formed by TM13 and TM14, in particular to satisfy PELDOR distance measurement L385R1. We added the following Figure to the Supporting Information:

Suppl. Fig. 5

Panel b: Uniqueness of rigid-body docking solution for SLC26 dimer. The reduced χ^2 of measured and calculated PELDOR signals for the three label sites is plotted as a function of the polar plane angle defining the orientation of the second protomer in the membrane plane under C_2 symmetry. Shown is the minimum χ^2 over the 10 monomer conformations in panel a. At an angle of 210 degrees (grey bar), all three PELDOR signals are reproduced, thereby defining the orientation taken for further analysis.

Please note that panel a is shown below.

Furthermore, a discussion about the uncertainty of the calculations should be added.

As is now shown in Suppl. Fig. 5b, the three labels quite uniquely triangulate the orientation of the two protomers in the SLC26 dimer. Since the values of the reduced χ^2 for different conformations in the membrane plane are very distinct, the uncertainty in the protein-protein docking procedure is about $\pm 5^\circ$, defined by the discrete step of the orientation scan. We now write:

“Due to the observed flexibility, we used several relaxed monomer conformations obtained at 110 ns intervals of MD for docking. For each conformation, a rigid-body search restricted by C2 symmetry with an axis normal to the membrane was performed and the rotation angle that showed the best overall fit with the PELDOR data was determined. Using this approach, we identified a candidate dimer structure based on a monomer conformation observed at 440 ns of MD and a polar plane angle of $210 \pm 5^\circ$ (Suppl. Fig. 5).”

2) If I understand correctly, docking was done on one conformation from the MD trajectories (a frame from 440 ns). Again, due to the limitation in docking, it would be important to see whether different solutions are predicted when different trajectories (or models) are docked. For example, one suggestion would be to cluster the entire trajectory, and then dock 3-5 relevant conformations and show some of the docking solutions in the supplementary material.

We thank the Reviewer for pointing out this crucial point of selecting a single conformation out of a 1000 ns MD simulation. In order to address this comment, we have visualized our analyzed conformations taken from several snapshots in the MD simulation (Suppl. Fig. 5). In particular, the PELDOR signal V367R1 was crucial in this analysis. We added the following figure to the Supporting Information:

Suppl. Fig. 5a: Selection of the SLC26 conformation to be used for rigid-body docking of the dimer from MD simulation of a membrane-embedded SLC26 monomer starting from the crystal structure. The consistency of potential SLC26 dimers with experimental PELDOR signals is quantified by the reduced χ^2 as a function of MD time for each spin-label position K353R1 (blue), V367R1 (red), and L385R1 (orange). The snapshot at 440 ns (grey bar) was taken for further analysis of the SLC26 dimer.

We now write: “Due to the observed flexibility, we used several relaxed monomer conformations obtained at 110 ns intervals of MD for docking. For each conformation, a rigid-body search restricted by C2 symmetry with an axis normal to the membrane was performed and the rotation angle that showed the best overall fit with the PELDOR data was determined. Using this approach, we identified a candidate dimer structure based on a monomer conformation observed at 440 ns of MD and a polar plane angle of $210 \pm 5^\circ$ (Suppl. Fig. 5).”

3) The dimeric model suggests that the interface between the protomers is uniquely small (350 Å²). I could imagine that a small change in the composition of the interface may have an effect on the dimer. Did the authors test whether some mutations break the complex formation (computationally and/or experimentally)? This would substantiate the prediction significantly.

We agree with Reviewer #3 that our determination of the SLC26Dg dimer interface has paved the way for future attempts to monomerize these proteins for subsequent functional characterization and the identification of potential extrinsic factors contributing to the dimerization. We have not performed computational or experimental mutagenesis studies towards this aim thus far. Similar studies have been performed for UraA (Yu et al. 2017) which showed that the monomer was well-folded, demonstrated by a binding assay, but no longer able to catalyze transport. Regrettably, we do not have any straightforward methodology (like a binding assay) to assess the folding quality of potential SLC26Dg monomers, due to the low affinity of SLC26Dg for its substrate fumarate. Consequently, we cannot discriminate between partly misfolded monomers and well-folded, but transport inactive versions. We are looking into alternative approaches to overcome this, but are currently not able to provide such data.

4) It will make it much easier to understand the manuscript, if Figure S4C-3 (with the residue visualization) will be integrated into Figure 2, and Figure S10E into Figure 5.

Following the suggestion of Reviewer #3 we have used Fig S10E to replace Fig 5a. This panel now more clearly illustrates the basis for the locking of SLC26Dg in the inward-open conformation. Fig. S4C-3 (now Suppl. Fig. 7C) visualizes the position of the rotamers in our dimer model. The information in this figure on the positions of the spin labels in the dimer model is in principle already present, but distributed over the main figures 1 and 3. We interpret the request of the Reviewer so that information on the labeled positions in the dimer model should be more readily available. For this, we have revised Fig. 3 in order to clearly indicate the positions of the labels in the context of the SLC26Dg dimer.

In addition, the more detailed description of the docking protocol in the supplementary material should be a part of the main text.

The detailed description in the Supplementary Material refers to an alternative docking approach based on the PELDOR distance distributions instead of the PELDOR time traces. We used both approaches in parallel as additional validation (comparison of both models in Suppl. Fig. 6). As our final model is based on the PELDOR time traces-based modelling approach, we prefer not to extend details on the other PELDOR distance distributions-based approach in the main text. Instead, we have provided more details on the time traces-based modelling in the main text (see below) and added two additional figure panels to further illustrate and clarify this approach.

We now write: “Due to the observed flexibility, we used several relaxed monomer conformations obtained at 110 ns intervals of MD for docking. For each conformation, a rigid-body search restricted by C2 symmetry with an axis normal to the membrane was performed and the rotation angle that showed the best overall fit with the PELDOR data was determined. Using this approach, we identified a candidate dimer structure based on a monomer conformation observed at 440 ns of MD and a polar plane angle of $210 \pm 5^\circ$ (Suppl. Fig. 5).”

Reviewers' Comments:

Reviewer #1:

Remarks to the Author:

I have no further comments to the authors... my major concerns were dealt with in the revised MS.

Reviewer #2:

Remarks to the Author:

The authors have carefully addressed all my concerns and rectified underlying issues. I fully support publication of the revised manuscript.

Reviewer #3:

Remarks to the Author:

The authors did excellent job addressing my comments as well as the comments made by the other reviewers.